# Greater Emotional Distress Due to Social Distancing and Greater Symptom Severity during the COVID-19 Pandemic in Individuals with Bipolar Disorder: A Multicenter Study in Austria, Germany, and Denmark

**DOI:** 10.3390/ijerph19137626

**Published:** 2022-06-22

**Authors:** Elena M. D. Schönthaler, Nina Dalkner, Michaela Ratzenhofer, Eva Fleischmann, Frederike T. Fellendorf, Susanne A. Bengesser, Armin Birner, Alexander Maget, Melanie Lenger, Martina Platzer, Robert Queissner, Adelina Tmava-Berisha, Christina Berndt, Julia Martini, Michael Bauer, Jon Dyg Sperling, Maj Vinberg, Eva Z. Reininghaus

**Affiliations:** 1Department of Psychiatry and Psychotherapeutic Medicine, Medical University Graz, 8036 Graz, Austria; elena.schoenthaler@medunigraz.at (E.M.D.S.); michaela.ratzenhofer@t-online.de (M.R.); eva.fleischmann@medunigraz.at (E.F.); frederike.fellendorf@medunigraz.at (F.T.F.); susanne.bengesser@medunigraz.at (S.A.B.); armin.birner@medunigraz.at (A.B.); alexander.maget@medunigraz.at (A.M.); melanie.lenger@medunigraz.at (M.L.); martina.platzer@medunigraz.at (M.P.); robert.queissner@medunigraz.at (R.Q.); adelina.tmava@medunigraz.at (A.T.-B.); eva.reininghaus@medunigraz.at (E.Z.R.); 2Department of Psychiatry & Psychotherapy, Faculty of Medicine, Carl Gustav Carus University Hospital, Technische Universität Dresden, 01307 Dresden, Germany; christina.berndt@uniklinikum-dresden.de (C.B.); julia.martini@ukdd.de (J.M.); michael.bauer@uniklinikum-dresden.de (M.B.); 3Psychiatric Research Unit, Psychiatric Centre North Zealand, 3400 Hillerød, Denmark; jonsperling@regionh.dk (J.D.S.); maj.vinberg@regionh.dk (M.V.); 4Department of Clinical Medicine, Faculty of Health and Medical Sciences, University of Copenhagen, 2200 Copenhagen, Denmark

**Keywords:** bipolar disorder, COVID-19, pandemic, social distancing, anxiety, symptom load

## Abstract

Throughout the COVID-19 pandemic, mental health of individuals with bipolar disorders (BD) is potentially more vulnerable, especially regarding COVID-19-related regulations and associated symptomatic changes. A multicentric online study was conducted in Austria, Germany, and Denmark during the COVID-19 pandemic. Overall, data from 494 participants were collected (203 individuals with BD, 291 healthy controls (HC)). Participants filled out questionnaires surveying emotional distress due to social distancing, fear of COVID-19, and the Brief Symptom Inventory-18 to assess symptom severity at four points of measurement between 2020 and 2021. General linear mixed models were calculated to determine the difference between the groups in these pandemic specific factors. Individuals with BD reported higher distress due to social distancing than HC, independently of measurement times. Fear of COVID-19 did not differ between groups; however, it was elevated in times of higher infection and mortality due to COVID-19. Individuals with BD reported higher psychiatric symptom severity than HC; however, symptom severity decreased throughout the measured time in the pandemic. Overall, individuals with BD experienced more distress due to the COVID-19 situation than HC. A supportive mental health system is thus recommended to ensure enhanced care, especially in times of strict COVID-19-related regulations.

## 1. Introduction

The outbreak of the novel severe acute respiratory syndrome coronavirus (SARS-CoV-2) resulting in the coronavirus disease (COVID-19) has led to a declared global health emergency due to its health consequences and lethality [1]. Because of the medical severity of the virus and the initial lack of treatment possibilities, several countries decided to enter sequential lockdowns to the extent that the majority of the worldwide population was instructed to stay at home [2]. Further, governmental restrictions with the purpose of containing the virus (e.g., keeping a distance from other people (“social distancing”), wearing face masks, not shaking hands) were introduced in most countries, thereby leading to a severe change and impairment of social life. Next to social consequences, the presence of COVID-19 has a medical [3], environmental, and economic impact [4,5,6]. Further, several cross-cultural studies revealed increased psychological impact due to COVID-19 and its consequences, reporting elevated psychological distress, anxiety, depression, emotional disturbances, panic, poor sleep quality, and lower resilience [7,8,9]. Although these studies show the severe impact of the pandemic on the mental health in the general population, those with a pre-existing psychiatric disorder were found to be even more affected by the pandemic, in some but not in all studies [10,11,12]. Apart from the higher risk of SARS-CoV-2 infections and worse COVID-19-related outcomes for individuals with psychiatric disorders [13,14,15], the uncertainty, social isolation, and economic problems caused by the pandemic [16] may lead to an intensification of psychiatric symptoms [10] and higher relapse rates of psychiatric disorders [17,18].

Concerning COVID-19, individuals affected by bipolar disorder (BD) seem to be an especially vulnerable group. BD is a recurrent affective disorder generally characterized by fluctuating episodes of depression and hypomania or mania [19], with stressful events being a well described risk factor that can lead to new affective episodes [20]. In the light of the pandemic as a stressful event, and the higher prevalence of somatic comorbidities in BD [21,22,23], it seems obvious that individuals with BD could be more affected by COVID-19. Further, they were shown to be more often infected with COVID-19, suffering from more severe illness courses and higher mortality due to the virus [14]. Moreover, circadian rhythm disturbances elicited by COVID-19-induced lockdowns or work in a home office could evoke or reinforce new episodes [24,25,26]. Indeed, an Australian study found that BD was associated with higher psychological distress during the COVID-19 pandemic, even in comparison to other psychiatric diseases [27]. Specifically, social distancing seems to have a negative impact on mood in individuals with BD, not least because of the disruptions in daily routines and structure, loss of income, job insecurity, and social support [28,29]. Further, COVID-19-induced lockdowns and subsequent social distancing expose individuals with BD to a higher risk of depressive relapses through the dysregulation of biological rhythms [17] and higher somatization symptoms [28]. Moreover, fears regarding COVID-19 may vary depending on lockdown regulations and could be associated with worse sleep quality in BD [30]. Finally, it was shown that the severity of bipolar symptoms changed due to COVID-19: One study indicated that individuals with BD reported more (hypo)manic symptoms due to COVID-19 restrictions [31]. However, other studies found that individuals with BD experienced less psychiatric symptoms during COVID-19 [12] or no change in symptoms at all [32]. Thus, more research is needed on the relationship between COVID-19-related psychological aspects and BD.

So far, few longitudinal data on the course of BD during the COVID-19 pandemic have been published. Furthermore, there is little evidence on differences in individuals with BD in several different countries during the pandemic, which is very much required given the fact that each country has different governmental strategies to prevent high infection and mortality rates due to COVID-19. However, the current literature suggests that fear of COVID-19 and severity of symptoms in BD vary depending on lockdown regulations and social distancing [31]. Thus, the current study examined emotional distress due to social distancing, fear of COVID-19, and severity of psychological symptoms in individuals with BD and healthy controls (HC) at four measurement times with different social distancing regulations (e.g., hard lockdown, mild lockdown, no restrictions) throughout the pandemic in three different European cities (Graz, Austria; Dresden, Germany; Copenhagen, Denmark).

The aim of the current study was to investigate to what extent individuals with BD and HC were affected by (1) emotional distress due to social distancing, (2) fear of COVID-19 and (3) symptom severity across four measurement times throughout the pandemic. Based on previous literature, we assumed that individuals with BD would show higher emotional distress due to social distancing during the measured time in the pandemic. Secondly, we hypothesized that fear of COVID-19 would be more elevated in individuals with BD during the measured time in the pandemic. Finally, we expected that the severity of psychological symptoms would increase during the measured time in the pandemic, especially in individuals with BD.

## 2. Materials and Methods

### 2.1. Participants

This study was sent out as an online survey at four measurement times during the ongoing COVID-19 pandemic to a pool of formerly collected email-addresses of individuals who had participated in other studies at each center and agreed to be contacted for further studies. Moreover, the study link was shared in social media platforms. In total, the data of 733 participants were collected in three study centers; however, 132 participants were excluded due to no or almost incomplete data entries, and 107 participants were excluded because they did not meet the inclusion criteria for HC of having no psychiatric diagnosis, no psychopharmaceutical medication, and no first-degree relative with a psychiatric diagnosis. Other than the abovementioned inclusion criteria for HC, participants of both groups fulfilled the inclusion criterion of being of legal age (≥18 years). Individuals with BD in Graz were diagnosed by psychologists and psychiatrists using the Structured Clinical Interview for DSM-IV-TR (SCID-I; [33]). This information was initially collected within the frame of other current and previous studies on BD of the Graz center. In Dresden and Copenhagen, they were invited from an outpatient clinic, thus self-reporting their diagnosis in the survey. All centers asked subjects of their formerly conducted studies, who provided their email-addresses, to participate in the current study. In total, a final sample size of 494 participants across all measurement times was obtained. Notably, not all subjects participated at each measurement time since some of the centers sent out the invitation only at measurement time 2 (Copenhagen) or measurement time 2–4 (Dresden). Moreover, it should be mentioned that the Danish center only collected data of bipolar individuals. Sample sizes at each measurement time can be found in Figure 1. 

All participants gave written, informed consent prior to participating in the study. This study was administered in accordance with the Declaration of Helsinki and was approved by the Ethics Committee of the Medical University of Graz covering all recruiting locations (EK-number: 25-335 ex 12/13).

### 2.2. Procedure

Data were collected via an online survey program (LimeSurvey 3.27.4). The Austrian center participated in all measurement times; however, Germany participated in measurement times 2–4 and Denmark in measurement time 2. The study link for the second and third measurement time was sent out to subjects who participated in either the baseline measurement and/or the first follow-up and provided us with their email-addresses. An overview of all measurement times, COVID-19-related restrictions and infection and mortality rates can be found in Table 1.

### 2.3. Material

This study was part of a large-scale study examining several other variables and overlapped with two other studies based on the baseline result [34,35]. These studies showed that the pandemic had a greater impact on the physical health of individuals with BD, in comparison to HC [35] and that emotional distress due to social distancing was related to more clinical symptoms (e.g., somatization, sleep quality) in individuals with BD than HC [34]. All questionnaires were administered in German (for Austria and Germany) or Danish. 

**Demographic data.** Relevant sociodemographic data were examined within a self- constructed questionnaire. For the current study, we examined the variables sex, age, education, and medication intake, the latter being a control variable for the diagnosis of BD.

**COVID-19 questionnaire.** Emotional distress due to social distancing was assessed with a self-constructed COVID-19 questionnaire. The following six items were rated on a five-point Likert scale, ranging from (0) = not at all to (4) = full commitment:

(1) I cope well with the social distancing and can occupy myself well.

(2–6) On a scale from 0–4, social distancing makes me feel lonely/bored/frustrated/hopeless/anxious.

Out of these significantly intercorrelated items (all *p*s < 0.01), a mean index for “emotional distress due to social distancing” was built by first reversing the scale for item 1 and then calculating the mean of all items. This index indicated sufficient internal consistency (Cronbach’s α = 0.75).

Fear of COVID-19 was measured within the same self-constructed questionnaire, comprising three questions. Subjects were asked to indicate their subjective fear on a 11-point Likert scale, ranging from (0) = “no fears” to (10) = “extremely high fear”:On a scale from 0–10, how strongly do you rate your concerns and fears about the coronavirus?On a scale from 0–10, how strongly do you rate your fear of contracting the coronavirus?On a scale from 0–10, how strongly do you rate your fear of infecting others with the coronavirus?

All items showed a highly significant intercorrelation (all *p*s < 0.01) and a mean index for COVID-19 fears was built, indicating sufficient internal consistency (Cronbach’s α = 0.81).

**Brief Symptom Inventory 18** (BSI-18; [36]). This short version of the Symptom-Checklist-90-Revised (SCL-90-R; [37]) assesses psychological symptoms throughout the last seven days. Psychological symptoms are measured with 18 items on three subscales (somatization, depression, anxiety) as well as the Global Severity Index (GSI) as a global measure of symptom load [36]. Items of the GSI indicated sufficient internal consistency (Cronbach’s α = 0.93).

### 2.4. Statistics

Due to the fact that some of the subjects did not participate in every measurement time, we used general linear mixed models (GLMMs) to determine the impact of BD on emotional distress due to social distancing, fear of COVID-19, and severity of psychological symptoms (GSI) across four measurement times. GLMMs are often used for longitudinal data analysis because they can be fitted using maximum likelihood methods that can handle varying numbers and timing of observations on subjects [38]. Preliminary analyses indicated that assumptions for conducting GLMMs were sufficiently met. Boxplots and Cook’s distance values revealed no influential outliers. Pearson-correlation analyses indicated no multicollinearity (all variance inflation factors > 1.01). Linearity, normal distribution of residuals and homoscedasticity were examined graphically with no deviations from assumptions found. Further, a priori bivariate Pearson-correlations revealed that there were no significant correlations between sex, age, and the dependent variables (all *p*s > 0.05), thus, there was no need to control for possible effects. We included time as the repeated measure and fixed effect in our GLMMs. We chose unstructured covariance patterns as these patterns model correlations and variances across all measurement times as they are, thus avoiding the possibility of specifying a wrong model [39]. The maximum likelihood method of estimation was applied due to its strong consistency [40]. For models showing a significant change of parameters across measurement times, post-hoc pairwise comparisons of the changes were performed with pairwise *t*-tests and corrected using the Bonferroni method. To control for possible effects, all analyses relevant to the hypotheses were conducted once without the Danish sample, as they only had the opportunity to participate once due to the one-time recruitment, and once with just the baseline sample to obtain information about those participants who participated from the very beginning and thus provide more detailed information. Here, GLMMs were used to observe the difference between BD and HC in social distancing, fear of COVID-19, and severity of psychological symptoms across measurement times. For all analyses, we used IBM SPSS Statistics (Version 27). All hypotheses were tested two-tailed at a significance level of α = 0.05. Data and analysis scripts can be accessed via https://osf.io/76gya/ (accessed on 10 June 2022).

## 3. Results

Relevant sample characteristics and participating centers across measurement times are displayed descriptively in Table 2.

For the first hypothesis, a GLMM examining the impact of BD and measurement time (as fixed and repeated measure) on emotional distress due to social distancing was calculated. Results indicated a significant main effect of BD on emotional distress due to social distancing (*F*(1, 547.17) = 25.80, *p* < 0.01, *η*^2^_p_ = 0.05). Further, there was no significant main effect of time (*F*(3, 310.38) = 0.74, *p* = 0.53, *η*^2^_p_ = 0.01) and no significant interaction between BD and time on emotional distress due to social distancing (*F*(3, 310.38) = 0.91, *p* = 0.44, η^2^_p_ = 0.01). Post-hoc pairwise Bonferroni-corrected comparisons revealed more emotional distress due to social distancing in BD than in healthy controls (HC; *M*_(BD)_ = 1.64, *SE*_(BD)_ = 0.51; *M*_(HC)_ = 1.27, *SE*_(HC)_ = 0.50, *F*(1, 547.17) = 25.90, *p* < 0.001, η^2^_p_ = 0.05).

For the second hypothesis, a GLMM investigating the effects of BD and measurement time (as fixed and repeated measure) on fear of COVID-19 was calculated. The results showed a statistically significant main effect of time (*F*(3, 279.45) = 8.21, *p* < 0.001, *η*^2^_p_ = 0.08), but no statistically significant effect of BD (*F*(1, 569.86) = 1.00, *p* = 0.32, *η*^2^_p_ = 0.00), and no statistically significant interaction between BD and time on fear of COVID-19 (*F*(3, 279.45) = 0.79, *p* = 0.50, *η*^2^_p_ = 0.01). Post-hoc pairwise Bonferroni-corrected comparisons showed a statistically significant decrease of COVID-19-related fear between the first and fourth measurement time (*M*_(t1)_ = 4.12, *SE*_(t1)_ = 0.20; *M*_(t4)_ = 3.27, *SE*_(t4)_ = 0.20), a statistically significant increase of COVID-19-related fear between the second and third measurement (*M*_(t2)_ = 3.73, *SE*_(t2)_ = 0.11; *M*_(t3)_ = 4.30, *SE*_(t3)_ = 0.17), and a statistically significant decrease between the third and fourth measurement time (*M*_(t3)_ = 4.30, *SE*_(t3)_ = 0.17; *M*_(t4)_ = 3.27, *SE*_(t4)_ = 0.20, *F*(3, 277.74) = 8.21, *p* < 0.001, *η*^2^_p_ = 0.08). All other comparisons showed no statistically significant differences (all *p*s > 0.17). 

Finally, a GLMM examining the impact of BD and measurement time (as fixed and repeated measure) on global severity index was computed. The results revealed statistically significant main effects of BD (*F*(1, 572.41) = 84.59, *p* < 0.001, *η*^2^_p_ = 0.13), and time (*F*(3, 236.81) = 8.31, *p* < 0.001, *η*^2^_p_ = 0.03), and a statistically significant interaction effect of BD and time on GSI (*F*(3, 236.81) = 4.47, *p* < 0.01, *η*^2^_p_ = 0.05). Post-hoc pairwise comparisons with Bonferroni adjustments showed that individuals with BD reported statistically significant higher GSI scores than HC (*M*_(BD)_ = 0.79, *SE*_(BD)_ = 0.04; *M*_(HC)_ = 0.45, *SE*_(HC)_ = 0.04, *F*(1, 572.41) = 84.59, *p* < 0.001, *η*^2^_p_ = 0.13). Moreover, post-hoc pairwise comparisons showed a significant decrease in the GSI score between measurement time 1 and 4 (*M*_(t1)_ = 0.63, *SE*_(t1)_ = 0.05; *M*_(t4)_ = 0.39, *SE*_(t4)_ = 0.05) and a statistically significant decrease between measurement time 2 and 4 (*M*_(t2)_ = 0.61, *SE*_(t2)_ = 0.03; *M*_(t4)_ = 0.39, *SE*_(t4)_ = 0.05). All other comparisons regarding the main effect of time on GSI were not statistically significant different (all *p*s > 0.09). Further post-hoc pairwise comparisons revealed a statistically significant decrease of the GSI score in individuals with BD between measurement time 1 and 3 (*M*_(t1)_ = 0.99, *SE*_(t1)_ = 0.07; *M*_(t3)_ = 0.69, *SE*_(t3)_ = 0.06) and measurement time 1 and 4 (*M*_(t1)_ = 0.99, *SE*_(t1)_ = 0.07; *M*_(t4)_ = 0.57, *SE*_(t4)_ = 0.06). Moreover, there was a significant difference between measurement time 2 and 3 (M_(t2)_ = 0.90, *SE*_(t2)_ = 0.04; *M*_(t3)_ = 0.69, *SE*_(t3)_ = 0.06) and measurement time 2 and 4 (*M*_(t2)_ = 0.90, *SE*_(t2)_ = 0.04; *M*_(t4)_ = 0.57, *SE*_(t4)_ = 0.06; *F*(3, 221.65) = 11.50, *p* < 0.001, *η*^2^_p_ = 0.13). All other post-hoc pairwise comparisons of GSI score across all measurement times were not statistically significant, either in the group of individuals with BD or in HC (all *p*s > 0.45). A visualization of the post-hoc comparisons can be seen in Figure 2. Notably, no data of the dependent variables before the pandemic was available, thus only changes in these variables throughout the pandemic can be described.

### Control Analyses

Results for the control analysis without the Danish sample followed the same pattern as the main analysis including all centers. Similar to the main analyses, we found a significant main effect of BD on emotional distress due to social distancing (*F*(1, 465.17) = 21.56, *p* < 0.001, *η*^2^_p_ = 0.04). Post-hoc pairwise Bonferroni-corrected comparisons revealed more emotional distress due to social distancing in BD than in HC (*M*_(BD)_ = 1.61, *SE*_(BD)_ = 0.54; *M*_(HC)_ = 1.27, *SE*_(HC)_ = 0.50, *F*(1, 465.17) = 251.56, *p* < 0.001, *η*^2^_p_ = 0.35). Further, we found a significant main effect of time on fear of COVID-19 (*F*(3, 270.29) = 8.98, *p* < 0.001, *η*^2^_p_ = 0.09). Post-hoc pairwise Bonferroni-corrected comparisons showed a statistically significant decrease of COVID-19-related fear between the first and fourth measurement time (*M*_(t1)_ = 4.07, *SE*_(t1)_ = 0.20; *M*_(t4)_ = 3.23, *SE*_(t4)_ = 0.20), a statistically significant increase of COVID-19-related fear between the second and third measurement (*M*_(t2)_ = 3.58, *SE*_(t2)_ = 0.13; *M*_(t3)_ = 4.26, *SE*_(t3)_ = 0.17), and a statistically significant decrease between the third and fourth measurement time (*M*_(t3)_ = 4.26, *SE*_(t3)_ = 0.17; *M*_(t4)_ = 3.23, *SE*_(t4)_ = 0.20, *F*(3, 269.27) = 8.98, *p* < 0.001, *η*^2^_p_ = 0.09). All other comparisons showed no statistically significant differences (all *p*s > 0.12). Moreover, results indicated significant main effects of BD (*F*(1, 490.78) = 68.99, *p* < 0.001, *η*^2^_p_ = 0.12), and time (*F*(3, 248.27) = 6.96, *p* < 0.001, *η*^2^_p_ = 0.08), and a statistically significant interaction effect of BD and time on GSI (*F*(3, 248.27) = 3.91, *p* < 0.01, *η*^2^_p_ = 0.05). Post-hoc pairwise comparisons with Bonferroni adjustments showed that individuals with BD reported statistically significant higher GSI scores than HC (*M*_(BD)_ = 0.73, *SE*_(BD)_ = 0.04; *M*_(HC)_ = 0.28, *SE*_(HC)_ = 0.04, *F*(1, 490.78) = 68.98, *p* < 0.001, *η*^2^_p_ = 0.12). Moreover, post-hoc pairwise comparisons showed a significant decrease in the GSI score between measurement time 1 and 4 (*M*_(t1)_ = 0.61, *SE*_(t1)_ = 0.05; *M*_(t4)_ = 0.37, *SE*_(t4)_ = 0.05) and a statistically significant decrease between measurement time 2 and 4 (*M*_(t2)_ = 0.55, *SE*_(t2)_ = 0.03; *M*_(t4)_ = 0.37, *SE*_(t4)_ = 0.05; *F*(3, 247.39) = 6.96, *p* < 0.001, *η*^2^_p_ = 0.08). All other comparisons regarding the main effect of time on GSI were not statistically significant different (all *p*s > 0.10). Further post-hoc pairwise comparisons revealed a statistically significant decrease of the GSI score in individuals with BD between measurement time 1 and 3 (*M*_(t1)_ = 0.94, *SE*_(t1)_ = 0.07; *M_(_*_t3)_ = 0.65, *SE*_(t3)_ = 0.06) and measurement time 1 and 4 (*M*_(t1)_ = 0.94, *SE*_(t1)_ = 0.07; *M*_(t4)_ = 0.53, *SE*_(t4)_ = 0.06). Moreover, there was a significant difference between measurement time 2 and 4 (*M*_(t2)_ = 0.78, *SE*_(t2)_ = 0.05; *M*_(t4)_ = 0.53, *SE*_(t4)_ = 0.06; *F*(3, 226.77) = 9.36, *p* < 0.001, *η*^2^_p_ = 0.11). However, deviating from the main analysis, no significant difference between measurement time 2 and 3 (*M*_(t2)_ = 0.78, *SE*_(t2)_ = 0.05; *M*_(t3)_ = 0.65, *SE*_(t3)_ = 0.06) was found. All other post-hoc pairwise comparisons of GSI score across all measurement times were not statistically significant, neither in the group of individuals with BD nor in HC (all *p*s > 0.19).

Results for the control analysis with just the baseline sample did not differ substantially from the main analysis including all centers. Similar to the main analyses, we found a significant main effect of BD on emotional distress due to social distancing (*F*(1, 89.42) = 14.77, *p* < 0.001, *η*^2^_p_ = 0.14). Post-hoc pairwise Bonferroni-corrected comparisons revealed more emotional distress due to social distancing in BD than in HC (*M*_(BD)_ = 1.70, *SE*_(BD)_ = 0.10; *M*_(HC)_ = 1.15, *SE*_(HC)_ = 0.10; *F*(1, 89.42) = 14.77, *p* < 0.001, *η*^2^_p_ = 0.14). Further, we found a significant main effect of time on fear of COVID-19 (*F*(3, 100.15) = 3.88, *p* < 0.05, *η*^2^_p_ = 0.10). Post-hoc pairwise Bonferroni-corrected comparisons showed a statistically significant decrease of COVID-19-related fear between the first and fourth measurement time (*M*_(t1)_ = 3.86, *SE*_(t1)_ = 0.23; *M*_(t4)_ = 2.86; *SE*_(t4)_ = 0.34, *F*(3, 100.10) = 3.88, *p* < 0.05, *η*^2^_p_ = 0.10), but, deviating from the results of the main analyses, no statistically significant increase of COVID-19-related fear between all other measurement times. All other comparisons showed no statistically significant differences (all *p*s > 0.16). Moreover, results indicated significant main effects of BD (*F*(1, 84.27) = 15.87, *p* < 0.001, *η*^2^_p_ = 0.16), and time (*F*(3, 91.60) = 8.90, *p* < 0.001, *η*^2^_p_ = 0.23), and a statistically significant interaction effect of BD and time on GSI (*F*(3, 91.60) = 5.17, *p* < 0.01, *η*^2^_p_ = 0.14). Post-hoc pairwise comparisons with Bonferroni adjustments showed that individuals with BD reported statistically significant higher GSI scores than HC (*M*_(BD)_ = 0.75, *SE*_(BD)_ = 0.10; *M*_(HC)_ = 0.23, *SE*_(HC)_ = 0.09, *F*(1, 84.27) = 15.87, *p* < 0.001, *η*^2^_p_ = 0.16). Moreover, post-hoc pairwise comparisons showed a significant decrease in the GSI score between measurement time 1 and 4 (*M*_(t1)_ = 0.61, *SE*_(t1)_ = 0.07; *M*_(t4)_ = 0.25, *SE*_(t4)_ = 0.09) and a statistically significant decrease between measurement time 2 and 4 (*M*_(t2)_ = 0.58, *SE*_(t2)_ = 0.08; *M*_(t4)_ = 0.25, *SE*_(t4)_ = 0.08) and 3 and 4 (*M*_(t3)_ = 0.52, *SE*_(t3)_ = 0.08; *M*_(t4)_ = 0.25, *SE*_(t4)_ = 0.08; *F*(3, 91.57) = 8.89, *p* < 0.001, *η*^2^_p_ = 0.23). All other comparisons regarding the main effect of time on GSI were not statistically significant different (all *p*s > 0.94). Further post-hoc pairwise comparisons revealed a statistically significant decrease of the GSI score in individuals with BD between measurement time 1 and 4 (*M*_(t1)_ = 0.98, *SE*_(t1)_ = 0.10; *M*_(t4)_ = 0.33, *SE*_(t4)_ = 0.13), 2 and 4 (*M*_(t2)_ = 0.90, *SE*_(t2)_ = 0.11; *M*_(t4)_ = 0.33, *SE*_(t4)_ = 0.133), and 3 and 4 (*M*_(t3)_ = 0.78, *SE*_(t3)_ = 0.12; *M*_(t4)_ = 0.33, *SE*_(t4)_ = 0.13; *F*(3, 91.91) = 11.72, *p* < 0.001, *η*^2^_p_ = 0.29). All other post-hoc pairwise comparisons of GSI score across all measurement times were not statistically significant, either in the group of individuals with BD or in HC (all *p*s > 0.30).

## 4. Discussion

This study set out to investigate the difference in emotional distress due to social distancing, fear of COVID-19, and severity of psychological symptoms between individuals with BD and HC across four measurement times with different legal regulations for social distancing throughout the COVID-19 pandemic in three European countries (Austria, Germany, Denmark). In summary, we found that individuals with BD experienced more emotional distress due to social distancing than HC. This effect was sustained throughout the period of the pandemic considered in this study and did not change at various measurement times. Further, it was found that fear of COVID-19 was not experienced differently by BD and HC. Notably, COVID-19-related fear increased statistically significantly in times of strict governmental restrictions, higher infection rates, and higher mortality among the general population in both groups. Finally, anxiety and depression symptoms, as measured with the BSI-18, were more pronounced in individuals with BD than in HC. However, severity of symptoms did differ across measurement times, with individuals with BD experiencing a significant decrease in symptom severity throughout the period of the pandemic considered in this study. For HC, no significant change in symptom severity was seen across all measurement times.

The finding of individuals with BD being more emotionally distressed by social distancing regulations than HC independently of varying measurement times throughout the recorded time in the pandemic was not in line with our expectation. However, this finding might be explained by the fact that individuals with BD generally tend to experience more stressful life events [41] and have a harder time recovering from stress [42] than HC. They also tend to use more maladaptive cognitive regulation of emotional events than HC [43,44], thus possibly experiencing more distress in general, independently of the current COVID-19 situation. However, our finding that individuals with BD were, in general, more emotionally distressed due to social distancing than HC is consistent with the current literature. Previous research was able to show that social distancing has a negative impact on psychological well-being in the general population [45,46], and specifically in individuals with BD, not least because of subsequent changes in routines, employment, and social support [28,29]. Specifically, social support is very important when it comes to avoiding recurrence of BD and striving for remission [47,48]; thus, it is obvious that withdrawal of social contacts leads to more emotional distress in BD [49]. Further, earlier research found that COVID-19-induced lockdowns and consequent social distancing result in a higher risk of depressive relapses [17] and more symptoms of somatization [28]. 

Further, our finding that fear of COVID-19 was equally represented in individuals with BD and HC, but differed across measurement times, was not in line with our previous assumption. Indeed, the current literature suggests that fear of COVID-19 leads to worse sleep quality [30] and is greater in times of strict lockdown measures in individuals with BD [31]. However, other authors found that fear of the virus was relatively equal in individuals with BD and HC and did not affect subjective life quality, mood lability, or changes in social rhythms or lifestyle factors [50]. Our finding of individuals with BD not differing from HC in terms of fear of COVID-19 supports this result. However, we also found that fear of COVID-19 was significantly greater at the third measurement time. This finding can be explained by the fact that this measurement was carried out in a period of high infection and mortality rates in European countries, with subsequent strict governmental regulations to contain virus spreading. 

Finally, our finding of individuals with BD experiencing higher symptom severity than HC across different measurement times during the recorded time of the COVID-19 pandemic supports our expectation. However, considering that individuals with BD generally experience more depressive or anxious symptoms than HC due to the disorder itself, this finding is not surprising. Interestingly, the severity of symptoms decreased throughout the period of the pandemic considered in this study in individuals with BD. A possible explanation of the reduction of symptom severity is the adaption to the new circumstances over the measured period. Specifically, the uncertainty regarding COVID-19 in the beginning of the pandemic might have triggered a higher intensity of depression and anxiety symptoms. With more time passing by, and more research being done on COVID-19, this uncertainty might have decreased, and with it the symptom load. Another explanation for this result is the gradually increasing therapy offered across the measurement times. In the beginning of the COVID-19 pandemic, therapy and mental health services were shut down to reduce the virus spreading. However, after a short period of adaption, online therapies and telemedicine were introduced, thus offering treatment possibilities for individuals with mental disorders, resulting in a subsequent decrease of symptom load. 

### Limitations

Our results should be interpreted with the following limitations in mind. First, the measurement of emotional distress due to social distancing and fear of COVID-19 did not follow a standardized inventory; however, the same items had been used in a previous cross-sectional pilot study (with participants in that study partially overlapping with ours [28]) and revealed a good internal consistency in the previous and current study. Secondly, this study was conducted online; thus, all responses were self-reported instead of externally and objectively rated. Moreover, we did not control for underlying personality factors such as neuroticism which could have had an effect on emotional distress, fear, and psychological symptoms, including depression. Third, our results are limited by the high amount of drop-outs between measurement time 2 and 3 in the group of HC in Graz. This may be explained by the fact that the third measurement time took place when COVID-19 was already an established phenomenon in people’s daily lives. Thus, participating in a study investigating the psychological consequences of COVID-19 for individuals with BD might not have been as interesting to HC anymore, leading to a high drop-out rate. Moreover, we recruited more HC than bipolar individuals via social media, and not all of them provided us with their email-addresses, which is why we could not contact the entire HC sample again for further measurements. Fourth, most of the BD group reported undergoing current treatment; however, we did not have this information for all participants. Moreover, BD and HC groups cannot be assessed for representativeness of target populations, thus possibly limiting generalizability. Finally, effect sizes for some of the results were rather small and should thus be interpreted accordingly.

## 5. Conclusions

This study is one of the first to measure emotional distress due to social distancing, fear of COVID-19, and severity of psychological symptoms in individuals with BD and HC in a longitudinal design during the COVID-19 pandemic. The study revealed that individuals with BD experienced more emotional distress due to social distancing than HC; however, fear of COVID-19 was equally high in both groups, and greater in times of strict governmental regulations and higher infection and mortality rates. Finally, the severity of psychological symptoms (anxiety, depression, somatization) was greater in individuals with BD than in HC. However, the symptom load decreased in individuals with BD throughout the measured time of the pandemic which could be a result of better mental health care in the course of the pandemic (e.g., online therapy). These findings point towards an increased vulnerability of individuals with BD, and possibly with all other psychiatric diseases, in times of the COVID-19 pandemic. Thus, a supportive mental health system is needed to ensure proper care and prevent possible negative consequences for individuals with mental disorders, especially in times of strict social distancing regulations.

## Figures and Tables

**Figure 1 ijerph-19-07626-f001:**
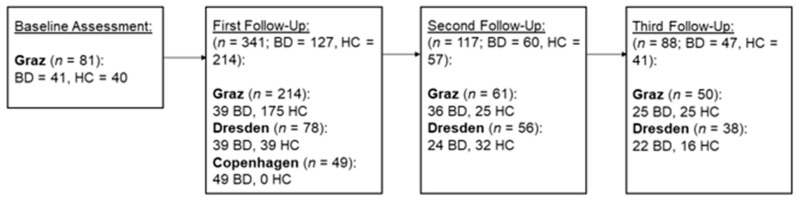
Overview of the sample sizes at each measurement time. Note. BD = Bipolar disorder, HC = Healthy controls.

**Figure 2 ijerph-19-07626-f002:**
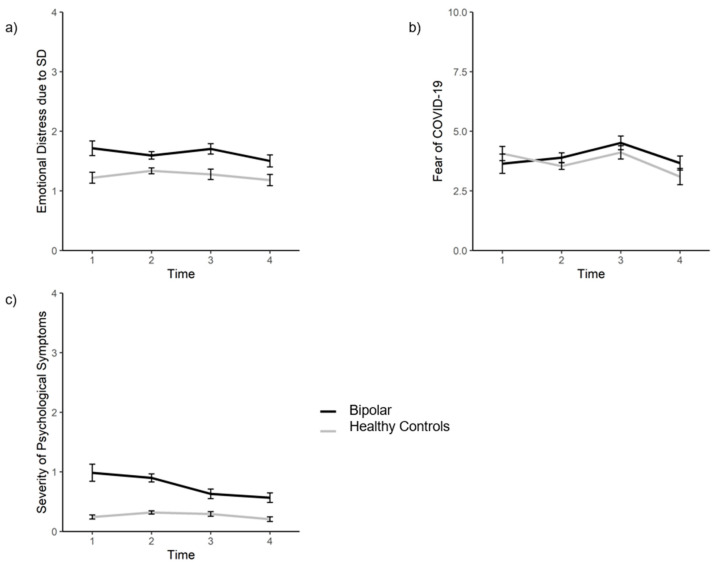
Emotional distress due to social distancing, fear of COVID-19, and severity of psychological symptoms scores across four measurement times in individuals with bipolar disorder (BD) and healthy controls (HC). Note. SD = Social Distancing. (**a**) Significant main effect BD, no significant main effect of time, no significant interaction between BD and time. (**b**) Significant main effect of time, but no significant main effect of BD or interaction between BD and time. (**c**) Significant main effects of BD and time, significant interaction effect of BD and time. Error bars indicate standard errors.

**Table 1 ijerph-19-07626-t001:** COVID-19-related restrictions, infection, and mortality rates for each measurement period in Austria, Germany, and Denmark.

	Austria			Germany			Denmark		
	Restrictions	Infection (per 100.000)	Mortality (per 100.000)	Restrictions	Infection (per 100.000)	Mortality (per 100.000)	Restrictions	Infection (per 100.000)	Mortality (per 100.000)
Baseline (Graz: 9–28 April 2020)	Social distancing; mandatory face masks in stores and public transport; all hotels, bars, and stores closed except for daily-needs stores; prohibition of events: closed schools and kindergartens; no travels or visits; no going outside without necessity	193	19	Social distancing (>1.50 m); mandatory face masks in stores, public transport; all hotels, bars, and stores closed expect for daily-needs stores; prohibition of events; closed schools and kindergartens; no travels or visits; no going outside without necessity	3.020	223	Social distancing; mandatory face masks; some shops remain open; prohibition of events; re-opening of schools and kindergartens; no travels or visits; going outside is possible without necessity	193	13
First Follow-Up (Graz: 5 May to 4 June 2020; Dresden: 17 June to 22 September 2020; Copenhagen: 15 June to 22 September)	Social distancing (>1 m); face masks in all public indoor areas; small stores and leisure parks re-open; events up to 10 people allowed; no travels or visits; gastronomy remains closed; school lessons and kindergarten in shifts with parted groups	34	3	Social distancing (>1.50 m); obligatory face masks in all public indoor areas; stores and some gastronomy re-opens under strict conditions; no travels or visits; school lessons and kindergartens in shifts with parted groups	1.224	23	Ease of rules; shops, bars, and restaurants re-open; schools and kindergartens re-open; universities and higher grades remain closed; restricted number of employees at workspace allowed	255	2
Second Follow-Up (Graz, Dresden: 5 November 2020 to 7 January 2021)	Social distancing; no going outside without necessity; mandatory face masks in all public indoor areas; re-closure of stores, universities, and secondary schools; gastronomy remains closed; schools and kindergartens in shifts	4.978	42	Social distancing; public gatherings with max. 10 people from two households; closure of leisure time activities and gastronomy; no travels and visits; schools and kindergartens remain open	33.947	657	Social distancing; local lockdowns; public gatherings limited to 10 people; bars and restaurants close at 22:00; mandatory face masks in indoor public areas; universities, secondary schools remain closed; indoor events prohibited	1533	21
Third Follow-Up (Graz, Dresden: 10 July to 8 September 2021)	Loosened social distancing; face masks only mandatory in supermarkets, public transport, and health care; schools, kindergartens, and universities fully re-opened; visit of bars, restaurants, hotels, and events and travelling possible if one is fully vaccinated, tested, or recovered from COVID-19	1.215	2	Loosened social distancing; face masks mandatory in stores, public transport, and health care; schools, kindergartens, and universities fully re-opened; visit of bars, restaurants, hotels, and events and travelling possible if one is fully vaccinated, tested, or recovered from COVID-19	2.960	4	All COVID-19-related restrictions are removed due to high vaccination rate	517	3

Note. Infection = Average of daily new COVID-19 cases at beginning and ending of measurement time. Mortality = Average of daily deaths due to COVID-19 at beginning and ending of measurement time. Grey colour indicates no participation of the center at this measurement time.

**Table 2 ijerph-19-07626-t002:** Demographic data for each measurement time.

	Variable	Group	
		BD	HC
Baseline Assessment	Total (*N*)	41	40
Center Graz (*n* = 81)	Age (*M*, *SD*)	49.95, 14.33	31.95, 7.86
(9–28 April 2020)	Sex (*n*, %)		
	Male	24 (58.54%)	10 (25%)
	Female	17 (41.46%)	30 (75%)
	Education (*n*, %)		
	GCSE/O levels	2 (5%)	0 (0%)
	Apprenticeship	13 (32.5%)	1 (2.5%)
	A- Levels	12 (30%)	9 (22.5%)
	Bachelor’s degree	2 (5%)	9 (22.5%)
	Master’s degree	10 (25%)	16 (40%)
	Doctoral level	1 (2.5%)	5 (12.5%)
First Follow-Up	Total (*N*)	127	214
Centers Graz (*n* = 214)	Age	40.95, 15.13	35.60, 12.90
(5 May to 4 June 2020)	Sex (*n*, %)		
	Male	84 (66.14%)	151 (70.56%)
Dresden (*n* = 78)	Female	43 (33.86%)	63 (29.44%)
(17 June to 22 September 2020)	Education		
Copenhagen (*n* = 49)	no formal education	5 (3.94%)	10 (4.67%)
(15 June to 22 September 2020)	GCSE/O levels	0 (0%)	0 (0%)
	Apprenticeship	28 (22.04%)	21 (9.81%)
	A- Levels	40 (31.50%)	32 (14.96%)
	Bachelor’s degree	25 (19.69%)	23 (10.75%)
	Master’s degree	26 (20.47%)	91 (42.52%)
	Doctoral level	3 (2.36%)	37 (17.29%)
Second Follow-Up	Total (*N*)	60	57
Centers Graz (*n* = 61),	Age	46.46, 15.21	36.63, 13.71
Dresden (*n* = 56)	Sex (*n*, %)		
(5 November 2020 to 7 January 2021)	Male	33 (55%)	49 (85.96%)
	Female	27 (45%)	8 (14.04%)
	Education		
	no formal education	1 (1.67%)	0 (0%)
	GCSE/O levels	2 (3.33%)	1 (1.75%)
	Apprenticeship	19 (31.67%)	4 (7.02%)
	A- Levels	20 (33.33%)	9 (15.79%)
	Bachelor’s degree	5 (8.33%)	12 (21.05%)
	Master’s degree	11 (18.33%)	24 (42.11%)
	Doctoral level	2 (3.33%)	7 (12.28%)
Third Follow-Up	Total (*N*)	47	41
Centers Graz (*n* = 50),	Age	48.17, 13.53	39.93, 12.69
Dresden (*n* = 38)	Sex (*n*, %)		
(10 July to 8 September 2021)	Male	22 (46.81%)	36 (87.80%)
	Female	25 (53.20%)	5 (12.20%)
	Education		
	no formal education	2 (4.26%)	0 (0%)
	GCSE/O levels	2 (4.26%)	1 (2.44%)
	Apprenticeship	15 (31.91%)	4 (9.76%)
	A- Levels	11 (23.40%)	6 (14.63%)
	Bachelor’s degree	2 (4.25%)	5 (12.20%)
	Master’s degree	13 (27.66%)	18 (43.90%)
	Doctoral level	2 (4.26%)	7 (17.07%)

Note. BD = Bipolar disorder, HC = Healthy controls, GCSE = General Certificate of Secondary Education.

## Data Availability

Data and analysis scripts can be accessed via https://osf.io/76gya/ accessed on 10 June 2022.

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
