# Peer review of "Greater Emotional Distress Due to Social Distancing and Greater Symptom Severity during the COVID-19 Pandemic in Individuals with Bipolar Disorder: A Multicenter Study in Austria, Germany, and Denmark"

_ijerph, 2022, doi:10.3390/ijerph19137626_

Round 1

Reviewer 1 Report

This paper describes analyses of data collected from patients with bipolar disorder (BD) and healthy controls (HC) in three European locations (Graz, Dresden and Copenhagen) over the early stages of the current SARS-CoV-2 pandemic from April 2020 to September 2021. The data analysis focusses on three outcome measures, the first two developed for the study: (1) emotional distress due to social distancing; (2) fear of COVID-19; and (3) scores on a short version of the SCL-90 covering 18 items of depression, anxiety and somatization. The main findings from general linear mixed models using four time points were: the BD group showed higher levels of emotional distress than the HC group across time points; there was no significant difference between the BD and HC groups for fear of COVID-19 but there were differences over time consistent with higher fear when infection and mortality rates were higher and restrictions were tighter; psychological symptoms of the BD group were significantly higher overall than for the HC group and declined over the four time points, with no such reduction for the HC group and a significant interaction term for group by time. The authors’ main conclusion is that the findings point to increased vulnerability of individuals with BD in times of the COVID-19 pandemic. Further, a supportive mental health system is needed to ensure proper care and prevent possible negative consequences for individuals with mental disorders, especially in times of strict social distancing regulations.

Overall, this manuscript is well written and presented and the Introduction and Discussion show a very good grasp of the relevant literature and the pertinent issues. In contrast there is a weakness in explaining why these particularly measures were selected for the present report and what relevance the findings have. The authors’ own conclusions come across as vague and do not appear to go beyond what most people would have imagined to be the case without the research. The description of the study design could also be made clearer. In particular, it is not obvious how the design relates to the statistical methodology. These and some more specific points are elaborated below.

1) The title of the paper does not seem accurate in two ways. First, it is a long stretch to describe the study as “longitudinal” (see below). Second, “higher symptom severity in times of the COVID-19 pandemic” in conjunction with longitudinal suggests a contrast that is not fulfilled. I presume this part of the title is based on the BSI symptom measure and derives from the higher scores of the BD group relative to HC group.

2)  I cannot recollect reading a paper which has been as honest as the present one in describing its original hypotheses and then going on to report results that mainly diverge from the stated expectations.

3) Reading Figure 1 along with Table 2 raises many issues concerned with the study design and procedures that are not explained in the text. What are the reasons behind the failure to recruit any HCs in Copenhagen? What is the explanation for the 175 HCs recruited in Graz dwindling to 25 by the second follow up? As a more general observation, there is no description of the recruitment procedure in the manuscript. Invitations were sent out but to whom? Other than the exclusion criteria for the HCs, what were the broader inclusion and exclusion criteria for the study?

4) Following on from the previous point, there is mention on page 7 (line 327) of “the high amount of drop-outs across the measurement times” but the reader cannot estimate the extent of drop-out based solely on numbers at individual time points. Were any new recruits included at the second and third follow-ups? Overall, what attrition occurred for the BD and HC groups? Given that Time 1 was restricted to participants in Graz, was there any value in including Time 1 measures in the data analyses? I found this very confusing.

5) Are the infection rates and mortality rates in Table 1 derived from national data or are they specific to the regions of the present study? It is a minor point given that no participants were recruited in Dresden at Time 1 but the figures imply a very high case fatality rate (7.4%) so would be worth checking.

6) In some parts of Table 2, percentages do not sum to 100%. Does Table 2 yield any information that the groups recruited are representative of the target populations?

7) I am sure few readers would interpret the current symptoms chart in Figure 2 to indicate a general improvement for the BD group due to the pandemic. Nevertheless, it seems worth mentioning that there is no means of knowing what the symptom levels would have been before the pandemic.

8) Something that struck me about Figure 2 and the Discussion sections relating to the three main outcome measures is that there is no sign of overlapping effects but rather three patterns for the three measures. I am not implying any criticism or suggesting any response to this; it is just an observation.

9) I have mentioned that the paper is well written. Occasional sentences seemed strange, however, and I was unsure if this was just due to the English expression. An instance was in the Abstract: “Overall, individuals with BD experienced more distress due to the COVID-19 situation as well as over time.” (line 27) I had no idea what the second part of this sentence meant (long term?). There were a number of references using phrases like “the whole pandemic” when the study did not (and cannot) cover the whole pandemic. In the limitations section there is a sentence: “Moreover, it should be noted that we did not control for distress in general.” Does this mean it should have been controlled for? If so, the measure from the BSI is available to use. Perhaps I have misunderstood the intention of the sentence.

Reviewer 2 Report

  Environmental Research and Public Health

Reviewer Comments

Title: Higher emotional distress due to social distancing and higher symptom severity 2 in times of the COVID-19 pandemic in individuals with bipolar disorder: a longi- 3 tudinal study in Austria, Germany, and Denmark

MS #: IJERPH-1720645

The purpose of this paper was to examine the impact of COVID-19 on the psychological stability of both Bipolar Disordered (BD) individuals and healthy controls (HC). Over time, it was found that levels of affective dysphoria increased for both groups, but the BD group had higher elevations than the HCs. Clearly, those who are emotionally vulnerable experienced more distress than HCs. A few comments.

1) It is not clear to me what this study adds that is new. BDs had higher levels of emotional distress and symptom experience over the entire duration of the study than controls. This is hardly surprising, given that one group is a clinical sample experiencing significant emotional problems. Those with BD have higher levels of affective dysphoria, poor coping skills, and problems with impulse control and self-regulation. These data show that these issues continued to exist, and may have been slightly exacerbated by the pandemic. When reading this paper, this is exactly the outcome I expected to find; it supports what is already known about BD individuals regarding personality, affective status, and the clinical dynamics of BD. So, I do not see what new ground is presented here.

2) In examining the graphs in Figure 2, some odd findings seem to be present. Regarding figure 2.1, Fear of Social Distancing, BDs are higher than HCs, which is what one would expect from individuals who are high on underlying levels of neuroticism because they tend to endorse more symptoms than those lower on neuroticism. However, from T1 to T2, BD levels seem to decrease while HC level increase. Then from T2 to T3 BD levels increase while HC levels decrease, then from T3 to T4, both groups decline, but the decline for the BD group seems steeper. Figure 2.2, regarding psychological symptoms, at T1 there is a very large difference between the two groups. I wonder what scores would look like if there was a pre-T1 assessment. Nonetheless, the BD group indicates on consistent lowering of symptom experiences while control go up slightly and then down. Finally, 2.3 and Fear of Covid, at T1 the BD group actually has a lower fear score than the controls, which then goes on to increase and peak at T3, while controls go down and then back up at T3. These are unexpected patterns and it is difficult to interpret what is going on here. I am concerned that these patterns may be more a function of the various data samples being analyzed rather than a reflection of some psychological process. The authors need to address this in the Discussion section. More on this below.

3) I am very uncomfortable with the recruitment of subjects and their usage in the various analyses. Each time period has a different set of subjects, in both groups, being employed for analysis. As such, it is not clear to me how these different samples may be different, both within group and between groups, on these scores initially. How many subjects were used consistently throughout the study? I am sure it cannot be more than the 81 people assessed at Baseline in Graz. So, as the composition of subjects being analyzed changes over time, to what extent do these intrinsic group characteristics are being reflected in the data rather than some substantive change in scores? Such “mix and match” sampling can have confounding effects on the data. How do I know that such confounding is not happening here? Also, there is no control for overall levels of neuroticism, which would be useful for comparing the two groups. I think it would be of value to examine the data, first off, by group. How do scores change for each group, independently, over time? What is the rate of change? I think it would be useful to demonstrate that the stress of the COVID pandemic had a more dysphoric impact (i.e., the rate of change in scores) on the BD sample than on the HC sample. It would also be useful to see how the adjustment to the pandemic positively impacted scores separately by group. Did it take longer for the BD group to return to baseline than the controls? This type of information may have more practical public healthy utility, by indicating when and for how long interventions need to be made. A second approach may include analyzing data separately by location. Following subjects at each location over time, may provide more interesting, and accurate, information. The patterns of results may be able to provide replications for the findings, because similar patterns were observed in different subjects at different locations. I believe that lumping everyone together adds too much noise to the results.

Round 2

Reviewer 1 Report

Many thanks to the authors for a very clear outline of how they have responded to previous comments. Overall, I am very happy with the changes made with the following exceptions.

Original point 3.

The revised manuscript now says “Other than the mentioned inclusion criteria for HC, all participants of the BD group fulfilled the inclusion criteria of being diagnosed with bipolar disorder.” This falls a long way short of a necessary description of the key BD group. Diagnosed when? By whom? No description is given as to how the BD individuals came to receive an invitation to participate. The numbers involved indicate they were not chance occurrences in amongst the email addresses that the researchers had available to them. Do the people in the BD group have current disorder? By what criteria? Were they currently in treatment for BD at the time of recruitment (or not)? Without a suitable description of this group, the reported findings have no generalizability.

Original point 4

The authors say in their response “Since we used general linear mixed models (GLMM), which are widely conceived to be the preferred method of analysis for repeated measurement designs when there are missing data due to dropouts, we were able to meaningfully include the baseline measure in Graz.” It is one thing to say this in a covering letter but a very different thing to demonstrate this to readers of the manuscript. I see that Figure 2 is essentially the same as in the original version of the paper. Each graph in the figure shows error bars (to indicate standard errors) only for Time 2 and Time 3. It is misleading to omit the error bars for Time 1 and Time 4. If the authors believe that these time points were “meaningfully” included in the GLMM analyses then they should add the error bars for Time 1 and Time 4 in the three graphs of Figure 2.

Original point 6

As the authors have acknowledged that the BD and HC groups cannot be assessed for representativeness of target populations, this needs to be added as a limitation in the appropriate section of the Discussion.

Round 3

Reviewer 1 Report

Thanks to the authors for addressing previous concerns with the manuscript.

Author Response

Thank you for your time and consideration.